# Selective Vision-Language Subspace Projection for Few-shot CLIP

## ABSTRACT

Vision-language models such as CLIP are capable of mapping the different modality data into a unified feature space, enabling zero/few-shot inference by measuring the similarity of given images and texts. However, most existing methods overlook modality gaps in CLIP's encoded features, which is shown as the text and image features lie far apart from each other, resulting in limited classification performance. To tackle this issue, we introduce a method called **S**elective Vision-Language **S**ubspace **P**rojection (**SSP**), which incorporates local image features and utilizes them as a bridge to enhance the alignment between image-text pairs. Specifically, our SSP framework comprises two parallel modules: a vision projector and a language projector. Both projectors utilize local image features to span the respective subspaces for image and texts, thereby projecting the image and text features into their respective subspaces to achieve alignment. Moreover, our approach entails only training-free matrix calculations and can be seamlessly integrated into advanced CLIP-based few-shot learning frameworks. Extensive experiments on 11 datasets have demonstrated SSP's superior text-image alignment capabilities, outperforming the state-of-the-art alignment methods. The code is available at: https://anonymous.4open.science/r/SSP-D3EC/main_our.py

## CCS CONCEPTS

• **Computing methodologies** → *Computer vision.*

## KEYWORDS

CLIP, Few-shot, Alignment, Subspace projection

**ACM Reference Format:**
Anonymous Author(s). 2024. Selective Vision-Language Subspace Projection for Few-shot CLIP. In *Proceedings of Make sure to enter the correct conference title from your rights confirmation email (Conference acronym 'XX)*. ACM, New York, NY, USA, 9 pages. https://doi.org/XXXXXXX.XXXXXXX

## 1 INTRODUCTION

The topic of pre-trained vision-language models (VLMs) [16, 21] has attracted significant research interest due to their exceptional performance in various multimodal tasks [12, 44, 50]. Among these, CLIP (Contrastive Language-Image Pretraining) [35] stands out in classification tasks, especially in zero/few-shot scenarios. For a given test image, CLIP computes the similarity between the image feature and the text features of all class labels, in the form of "a

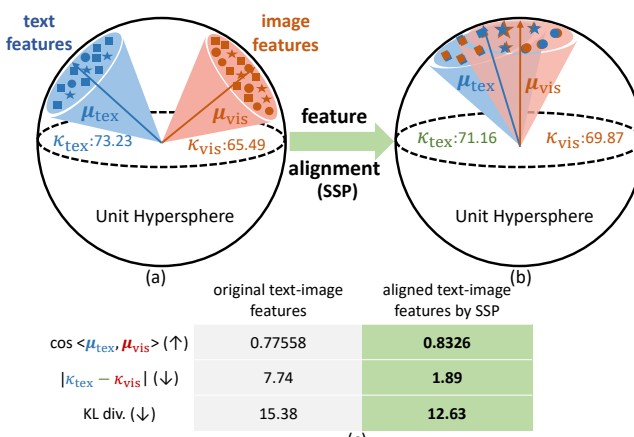

**Figure 1: A illustration of modality gaps conducted on ImageNet [5] with ViT-B/32. (a) Text and image features from CLIP lie in different cones. (b) Text and image features aligned by SSP almost line in the same cone. (c) Comparisons of distribution metrics for CLIP and SSP.**

photo of a [class name]. ". It then assigns the class with the highest similarity as the predicted label.

Recently, prompt tuning [52, 53] and adapter tuning [9, 41, 51] techniques have been applied in CLIP to further explore its generalization ability in few-shot tasks. These methods add learnable embeddings or adapter layers to either the visual encoder or textual encoder of CLIP. While they have improved the performance, they overlook the modality gaps existed in multi-modal models. As discussed in [24, 32, 40] and illustrated in Figure 1(a), modality gaps refer to different modality embeddings, *e.g.*, CLIP's encoded text features (pink cone) and image features (blue cone), are located in two separated regions in a hypersphere. To better understand these gaps, we use von Mises-Fisher (vMF) distribution [1, 6] to fit the text and image distributions, considering both modality features are normalized to unit length and lie on a unit hypersphere, where the parameters $\mu$ and $\kappa$ in vMF are analogous to the mean and standard deviation of Gaussian distribution, respectively (details of vMF formalization can be found in the Supplementary Material). Figure 1(c) summarizes the results, where the original text and image features show different distributions, *e.g.*, a large angle between $\mu_{\text{tex}}$ and $\mu_{\text{vis}}$, and a large $\ell_1$-norm between $\kappa_{\text{tex}}$ and $\kappa_{\text{vis}}$. This phenomenon is unexpected, as paired text-image features are optimized during the pre-training stage to be closely located to each other while being separated from other paired image-text features. To further investigate these gaps, we visualize the activation maps by calculating the similarity maps between the text features and local image features (feature map) in Figure 2, the second row shows that the text features focus on the unrelated regions, not just the foreground objects. This also demonstrates the gaps between the text and image features. In general, several reasons can cause

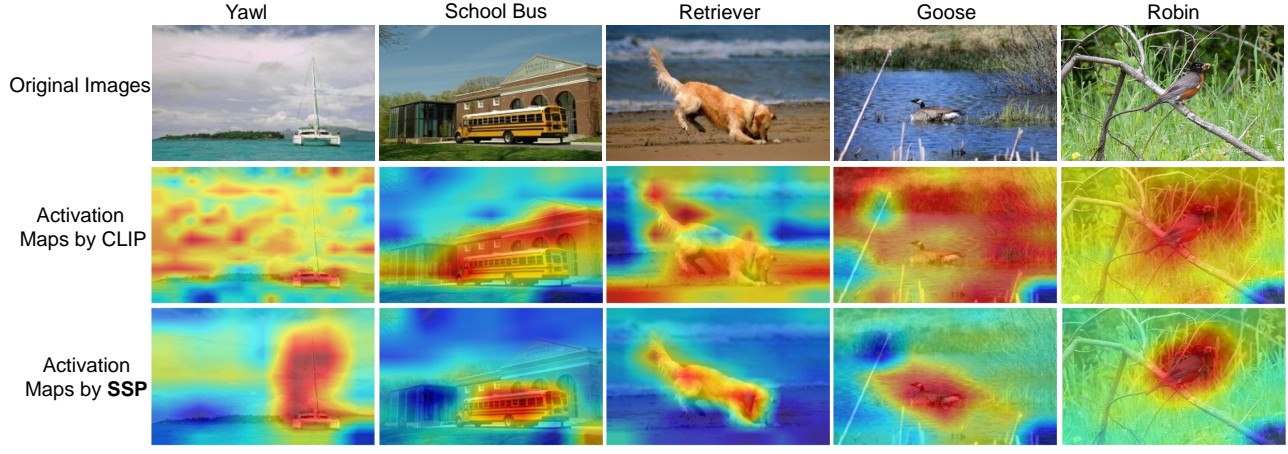

Figure 2: **Comparisons of class activations maps, where the CLIP's encoded features may concentrate on opposite or noisy regions as discussed in [23], while our SSP-aligned features primarily foreground objects.**

the modality gaps, broadly categorized as follows: (1) differences in downstream dataset distribution from the training data distribution [48], (2) different random model initializations cause the different feature cones [24], (3) CLIP model does not incorporate the fine-grained relationship between text tokens and image patches [8, 25], while these works on analyzing the modality gaps, they do not concentrate on CLIP's few-shot generalization capability.

Based on the observation and analysis, we propose a training-free method called **S**elective Vision-Language **S**ubspace **P**rojection (**SSP**), that leverages the local image features as a bridge to align image and text features. Our method aims at reducing the modality gap so that the paired text-image features are no longer farther apart from each other, as shown in Figure1(b) (the pink cone and blue cone are closer). To be specific, our SSP consists of two parallel modules, namely, a vision projector and a language projector. In the vision projector module, we utilize the regions of local image features that are similar to the image features to create a unified vision subspace due to the common structural and textural elements present in images, and then all image features are projected onto this vision subspace to achieve alignment. The language projector module operates in a similar manner, where we use local image features to construct the language subspace for each class. Subsequently, text features extracted from CLIP are aligned by projecting them onto their respective language subspaces. During the inference stage, all projected features are employed to classify the test image. The effectiveness of SSP can be seen from the metric results in Figure 1(c), by narrowing the modality gap, the text-image features aligned by SSP exhibit a higher cosine similarity, *e.g.*, 0.7755 vs. 0.8325, as well as the smaller difference of $\kappa$, and they also obtain a closer distribution distance by comparing the KL divergence. Besides, the visualized activation maps in the last row of Figure 2 show that text features processed by SSP can mainly focus on the foreground object. Overall, our SSP method only involves training-free matrix calculations, which can enhance the CLIP's capability in few-shot scenarios by improving the alignment between the paired text-image features.

The main contributions of our SSP are summarized as follows:

(1) We propose a training-free method SSP to reduce the modality gaps in CLIP's encoded features, which improves the CLIP's few-shot generalization ability.

(2) We design vision and language projectors, which leverage regions of the local image features to align the image and text features via subspace projection.

(3) Our SSP is flexible and can be applied to various CLIP-based methods, improving their performance across diverse benchmarks, even in state-of-the-art methods, *e.g.*, an average accuracy improvement of 0.63% over APE [55] in 16-shot.

## 2 RELATED WORK

In this section, we first introduce the Vsion-Language Models. Then we present CLIP with prompt tuning and adapter tuning in the few-shot scenario and compare our SSP with related methods.

## 2.1 VLMs Pre-training

VLMs have gained significant advances in recent years [26, 49]. These models bridge the modalities of vision and language and are typically pre-trained on a large dataset. With image-text pairs, VLMs use an image encoder and a text encoder to extract image and text features, and then learn the vision-language correlation by using certain pre-training objectives. Moreover, a range of VLMs, including CLIP (Contrastive Language-Image Pretraining) [23, 35], CoCa (Contrastive Cross-modal Learning) [45], and BLIP (Bidirectional Learning of Image and Text Priors) [20], can be leveraged for various downstream tasks, such as object recognition [51, 54], object detection [10], and image captioning [2, 30]. For instance, CLIP is pre-trained on a vast dataset of web-based image-text pairs and learns to align their representations through contrastive loss, which enables it to recognize unseen data by matching the embeddings of any given images and texts. In this paper, we aim to use CLIP to address few-shot classification problems.

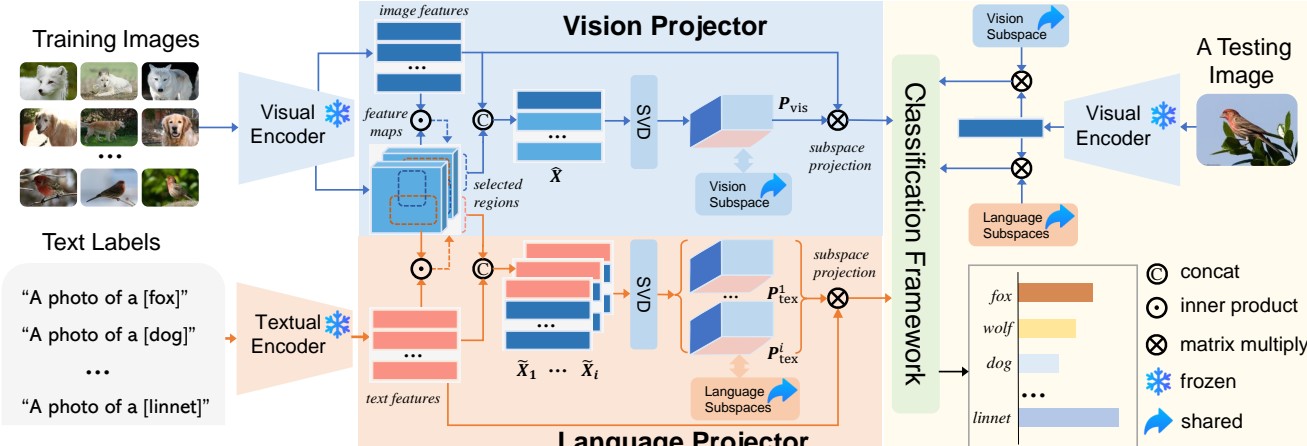

Figure 3: **The overview of our approach. The training images and extended labels are sent to the frozen visual encoder and textual encoder to extract features, respectively. Subsequently, the related local image features (features maps) are employed to construct the vision subspace and language subspaces, which are performed to align the extracted image and text features through subspace projection. Finally, a projected testing feature along with projected training features are inputted into the classification framework to predict results.**

## 2.2 Prompt Tuning of VLMs

The concept of prompt tuning is initially introduced in the natural language processing area, which refers to utilizing a fixed part of the text input as learnable embeddings and fine-tuning its parameters based on the downstream task data. CoOp [53] uses learnable word embeddings to generate context prompts automatically, eliminating the need for manual prompt templates. CoCoOp [52] proposed a meta network to learn image features that served as conditions added to prompt embeddings to further enhance the model's generalization ability. ProDA [27] aims to learn the distributions of the prompt embedding. ProGrad [54] used zero-shot prediction results to direct the model gradient update, preventing conflicts between few-shot models and general knowledge while mitigating overfitting issues. However, the above-mentioned methods, although they enhance the classification performance of CLIP in few-shot scenarios, involve the introduction of learnable parameters and increase training consumption. In contrast, our SSP only involves matrix calculation and does not introduce any learnable parameters.

## 2.3 Adapter Tuning of VLMs

Adapter tuning techniques are applied in VLMs [9, 41, 51, 55] to enhance downstream generalization ability by freezing the original model parameters and only updating the parameters of added adapter module. CLIP-Adapter [9] utilized a fully connected layer to adapt the features outputted from the frozen CLIP. Tip-Adapter [46] leverages a cache model to measure the relations between image and text features to construct a classifier based on few-shot training data. APE [55] represents an enhanced version of Tip-adapter, selecting the most discriminative feature channels via statistical analysis. SuS-X [41] relies on the category names from the training set to generate image samples based on stable diffusion [37]. Cross-Modal Adapter (CMA) [17] achieves cross-modal interaction by

sharing adapter weights between two modalities. While the aforementioned methods adapt text-image features for the downstream tasks, they do not account for their modality gaps. Our SSP method strives to reduce the gaps to better align the paired text-image features and can be integrated into the aforementioned methods.

## 3 METHODOLOGY

In this part, we provide a detailed explanation of our methodology. Firstly, we offer a brief overview of the zero-shot inference ability of CLIP in Section 3.1. Then, we delve into the specifics of our vision projector, discussed in Section 3.2, and language projector discussed in Section 3.3. Following that, we present the classification process of SSP in Section 3.4. Lastly, we conduct a comparative analysis between our SSP approach and related methods in Section 3.5.

## 3.1 Preliminaries

The pre-trained CLIP model can be adapted to the downstream tasks, *i.e.*, zero-shot classification. This process involves extending the "[class name]" to the template "a photo of a [class name]". Subsequently, the image feature $f \in \mathbb{R}^d$ and the class-extended text feature $t_i \in \mathbb{R}^d$ are extracted from CLIP's visual encoder and textual encoder, respectively. The classification is determined by the cosine similarity $\langle f, t_i \rangle$:

$$p(t_y|f) = \frac{\exp(\langle f, t_y \rangle)}{\sum_{i=1}^{N} \exp(\langle f, t_i \rangle)}, \qquad (1)$$

where $N$ represents the number of classes. In the few-shot setting of CLIP, there are given $N$ classes, each class containing $K$-shot training images. In our method, apart from the encoded image feature $f$ and the text feature $t$, we also exploit local image features $x \in \mathbb{R}^{hw \times d}$, where $h$ and $w$ represent the height and width of the local image feature, respectively. The overview of our method is depicted in Figure 3. We select local image features that exhibit

strong correlations with both image and text features, respectively, as determined by cosine similarity. Then we utilize these selected local features to construct the vision subapace and language subspaces, respectively. In the inference stage, the test image feature is projected into the vision and language subspaces, respectively, and then the projected training text-image features and the projected test features are sent to the classifier to predict the result.

## 3.2 Vision Projector

In the vision projector module, the cosine similarity between the image feature and the local image features is calculated for each sample in the training set as follows:

$$s_{i,j} = f_{i,j} \cdot x_{i,j}^{\mathrm{T}}, \ i \in [1, N], \ j \in [1, K]. \tag{2}$$

Here, $f_{i,j}$ is the image feature of $j$-th sample from class $i$, and $x_{i,j}$ denotes the corresponding local image features. $K$ indicates the number of samples within each class. $s_{i,j} \in \mathbb{R}^{hw}$ denotes the vector of similarity scores, with each element indicating the correlation between the image feature and the local region feature. Previous studies [23, 29] have demonstrated that local image features play a crucial role in capturing object and semantic information for vision and language, respectively. Consequently, a higher value in $s_{i,j}$ indicates that this local region contains more discriminative information for correct classification. Conversely, a smaller value suggests that the region includes less useful or irrelevant information for the corresponding class. Based on these insights, we utilize regions with top-$Q$ largest values to construct the vision subspace:

$$\hat{x}_{i,j} = x_{i,j}[n,:] \in \mathbb{R}^{Q \times d}, \ n = \{n_1, n_2, \cdots, n_Q\}, \tag{3}$$

where $Q$ is a hyper-parameter validated in Section 4.2.1, and $n = \{n_1, n_2, \cdots, n_Q\}$ denotes the indices set of top-$Q$ largest values in $s_{i,j}$. By aggregating these local features across all training samples and concatenating them with $f_{i,j}$, we obtain:

$$\hat{X} = [f_{1,1}, \hat{x}_{1,1}, \cdots, f_{i,j}, \hat{x}_{i,j}, \cdots, f_{N,K}, \hat{x}_{N,K}]^{\mathrm{T}}$$
$$\in \mathbb{R}^{((N+1)K \times Q) \times d}. \tag{4}$$

$\hat{X}$ comprises the informative features for vision patterns, and we decompose the $\hat{X}$ by Singular Value Decomposition (SVD):

$$U_{\mathrm{vis}} \Sigma_{\mathrm{vis}} V_{\mathrm{vis}}^{\mathrm{T}} = \mathrm{SVD}(\hat{X}), \tag{5}$$

where $U_{\mathrm{vis}} \in \mathbb{R}^{((N+1)K \times Q) \times ((N+1)K \times Q)}$ and $V_{\mathrm{vis}} \in \mathbb{R}^{d \times d}$ is the left singular vectors and right singular vectors, respectively, and they are corresponded to the singular values $\Sigma_{\mathrm{vis}} \in \mathbb{R}^{((N+1)K \times Q) \times d}$ sorted in the descending order. We employ the principal vectors of $V_{\mathrm{vis}}$, denoted as $\hat{V}_{\mathrm{vis}}$, to span the visual subspace [38], and the corresponding projection matrix is calculated as [11, 47]:

$$P_{\mathrm{vis}} = \hat{V}_{\mathrm{vis}} \hat{V}_{\mathrm{vis}}^{\mathrm{T}}, \tag{6}$$

where $P_{\mathrm{vis}} = P_{\mathrm{vis}}^{\mathrm{T}} \in \mathbb{R}^{d \times d}$. Depending on the constructed vision subspace and projection matrix, we align the image features via subspace projection:

$$\hat{F} = P_{\mathrm{vis}} F, \tag{7}$$

where $F = [f_{1,1}, \cdots, f_{N,K}] \in \mathbb{R}^{d \times NK}$ refers to all training image features, and the projected image features $\hat{F}$ are then utilized in the classification process.

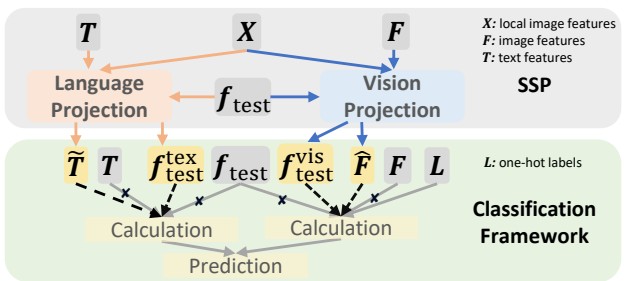

Figure 4: **The main differences between our SSP and other adapter-based methods [32, 51, 55].**

## 3.3 Language Projector

The language projector module has the same effects as the vision projector, and they exhibit a similar procedure. The cosine similarity between the text feature and local image features is calculated as:

$$z_{i,j} = t_i \cdot x_{i,j}^{\mathrm{T}}, \ i \in [1, N], \ j \in [1, K], \tag{8}$$

where $t_i$ is the text feature of $i$-th class, and $x_{i,j}$ denote the local image features of $j$-th image for class $i$. $z_{i,j} \in \mathbb{R}^{hw}$ stores the similarity scores between the text feature and the local image feature. The larger values in $z_{i,j}$ imply that these local regions exhibit highly semantic with $t_i$, and we select regions with top-$C$ largest values to construct language subspace:

$$\tilde{x}_{i,j} = x_{i,j}[m,:], \ m = \{m_1, m_2, \cdots, m_C\} \in \mathbb{R}^{C \times d}, \tag{9}$$

where $m = \{m_1, m_2, \cdots, m_C\}$ represents indices of top-$C$ largest values in $z_{i,j}$. For each text feature $t_i$, there are $K$ local image features belonging to that class. Consequently, we can gather a total of $K \times C + 1$ features, denoted as $\tilde{X}_i = [t_i, \tilde{x}_{i,1}, \cdots, \tilde{x}_{i,K}]^{\mathrm{T}} \in \mathbb{R}^{((K \times C+1) \times d)}$ for each class. In contrast to a unified vision subspace, we construct a language subspace for each class based on $\{\tilde{X}_i\}_{i=1}^N$ via SVD, as shown below:

$$U_{\mathrm{tex}}^i \Sigma_{\mathrm{tex}}^i (V_{\mathrm{tex}}^i)^{\mathrm{T}} = \mathrm{SVD}(\tilde{X}_i), \ i \in [1, N], \tag{10}$$

where $U_{\mathrm{tex}}^i$ and $V_{\mathrm{tex}}^i$ are the left and right singular vectors, respectively, corresponding to the singular values $\Sigma_{\mathrm{tex}}^i$ sorted in descending order. $\tilde{V}_{\mathrm{tex}}^i$ denotes the primary singular vectors of $V_{\mathrm{tex}}^i$, which forms the basis for the $i$-th language subspace. The corresponding projection matrix is denoted as $P_{\mathrm{tex}}^i = \tilde{V}_{\mathrm{tex}}^i (\tilde{V}_{\mathrm{tex}}^i)^{\mathrm{T}} \in \mathbf{R}^{d \times d}$. Subsequently, the text features are projected by their corresponding projection matrix as follows:

$$\tilde{T} = [\tilde{t}_1, \cdots, \tilde{t}_i, \cdots, \tilde{t}_N]^{\mathrm{T}} \in \mathbb{R}^{N \times d},$$
$$= [P_{\mathrm{tex}}^1 t_1, \cdots, P_{\mathrm{tex}}^i t_i, \cdots, P_{\mathrm{tex}}^N t_N]. \tag{11}$$

The projected text features $\tilde{T}$ and the projected image features $\hat{F}$ belonging to the same class lie closely to each other, as depicted in Figure 1 (b), Thereby, these paired text-image features are utilized for classifying test images during the inference stage.

## 3.4 Classification Process

Our SSP doesn't involve the parameters training process and can seamlessly build on top of existing adapter-based methods, such as LFA[32], Tip[51], and APE[55], to achieve improved classification

results, as summarized in Table 2 and Table 3. Specifically, given a test image $f_{\text{test}} \in \mathbb{R}^d$, it's projected into the vision subspace and language subspace, respectively. The vision subspace projection process is straightforwardly calculated as:

$$f_{\text{test}}^{\text{vis}} = P_{\text{vis}}f_{\text{test}}, \tag{12}$$

where $f_{\text{test}}^{\text{vis}}$ denotes the projected feature. However, for the language subspace projection, we encounter a challenge since we lack categorical information to determine the appropriate language projection matrix to use. To address this, we rely on projection theory [47] to choose the language projection matrix based on minimizing the $\ell_2$ norm of the orthogonal projected features:

$$\arg\min_i \; ||(I - P_{\text{tex}}^i)f_{\text{test}}||_2^2, \; i \in [1, N] \tag{13}$$
$$f_{\text{test}}^{\text{tex}} = P_{\text{tex}}^i f_{\text{test}}.$$

If the $f_{\text{test}}$ lies in the $i$-th language subspace entirely, the projection operation doesn't change the norm of the test feature, *i.e.*, $f_{\text{test}} = P_{\text{tex}}^i f_{\text{test}}$. After aligning the test feature via our vision and language subspace projection. The aligned test features along with the aligned image and text features are inputted into the various classification frameworks the get the prediction. For example, if we choose the LFA [32] as the classifier, we calculate the transformation matrix $W$ based on aligned image features $\hat{F}$ and text features $\tilde{T}$. We then use $f_{\text{test}}^{\text{tex}}$ to compute the final classification logits, given by $\tilde{T} \cdot W \cdot f_{\text{test}}^{\text{tex}}$. If we choose the Tip [51] or APE [55] as the classification frameworks, we utilize both $f_{\text{test}}^{\text{vis}}$ and $f_{\text{test}}^{\text{tex}}$ to calculate the prediction results, which are given by $\hat{F} \cdot f_{\text{test}}^{\text{vis}} + \tilde{T} \cdot f_{\text{test}}^{\text{vis}}$.

## 3.5 Analysis of SSP

Our method focuses on aligning the image and text features and can be easily built on top of various classifiers as described above. This sets our SSP approach apart from adapter-based methods such as Tip[51], APE[55], and LFA[32]. Although these methods operate on adapting the feature encoded from CLIP, they can also be viewed as classification frameworks due to their direct calculations between the training and testing samples. As shown in Figure 4, the adapter-based methods are depicted in the green box at the bottom. They employ various techniques to calculate the relationship among $T$, $F$, $L$, and the test sample $f_{\text{test}}$ to achieve classification, where $L$ signifies the one-hot labels for $F$. In contrast, our method, illustrated in the gray section at the top, leverages local image features $X$ as a bridge to align the image features, text features, and test features, resulting in $\tilde{T}$, $\hat{F}$, $f_{\text{test}}^{\text{vis}}$, and $f_{\text{test}}^{\text{tex}}$. These aligned features are substituted for the original ones in the computations to derive the classification results.

## 4 EXPERIMENTS

In this section, we first describe the experimental settings, and then we conduct ablation studies to analyze the effectiveness of different components within our method. Additionally, we compare the performance of our approach with other state-of-the-art (SOTA) methods. Finally, we evaluate the robustness of our method by testing it on out-of-distribution datasets. The experiments aim to address the following research questions (**RQ**):

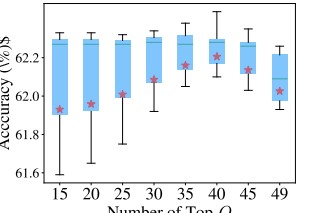 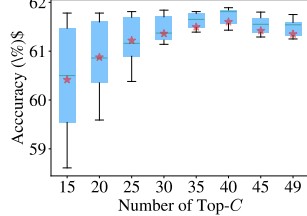

(a) **Selection of Regions for Vision Projector.**  (b) **Selection of Regions for Language Projector.**

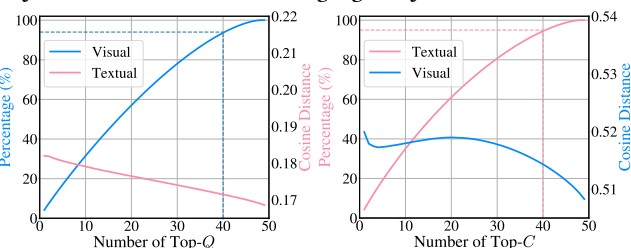

(c) **Trend of similarity between text features and local features.**  (d) **Trend of similarity between image features and local features.**

Figure 5: **Comparison of classification accuracy and percentage measurements by varying the number of selected local image features under the 16-shot setting: (a) Classification results using only vision subspace projection. (b) Classification results using only language subspace projection. (c) Relationship between text features and local image features selected by image features. (d)Relationship between image features and local image features selected by text features.**

**RQ1:** What is the optimal number of selected regions for image features and text features, respectively?
**RQ2:** How many principal singular vectors should be used to construct the vision projection matrix and language projection matrix, respectively?
**RQ3:** What is the impact of the projection operation on the classification results?
**RQ4:** How does the performance of our approach compare to that of state-of-the-art methods?
**RQ5:** What is the generalization ability of our method?
**RQ6:** How does the computational efficiency of our method compare to that of other methods?
By addressing these research questions systematically, we aim to provide a comprehensive evaluation of the effectiveness and robustness of our proposed methodology.

## 4.1 Experimental Settings

**Datasets.** In our experimental evaluation, we conducted experiments on 11 widely-used image classification benchmarks, covering a diverse range of object, scene, texture, and fine-grained categories. The datasets used in our experiments comprise ImageNet [5], Caltech101 [7], DTD [4], EuroSAT [13], FGVCAircraft [28], Flowers102 [31], Food101 [3], OxfordPets [33], StanfordCars [19], SUN397 [43], and UCF101 [39]. Additionally, we adopted variants of ImageNet, namely, ImageNetV2 [36], ImageNet-Sketch [42], ImageNet-A[15],



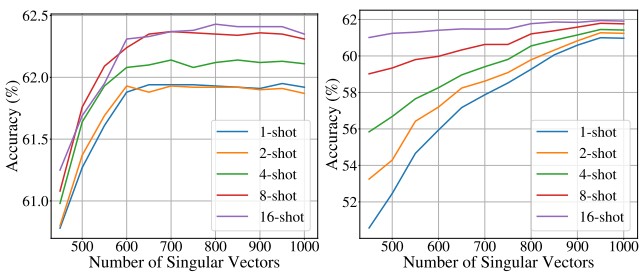

(a) **Vision Subspace Projection.** (b) **Language Subspace Projection.**

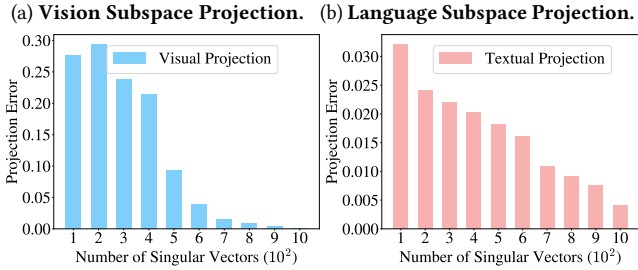

(c) **Vision Projection Analysis** (d) **Language Projection Analysis**

Figure 6: **The accuracy (%) of the classifiers and projection errors with varying the number of singular vectors: (a) The vision projection matrix construction for the different number of shots, (b) The language projection matrix construction with the different number shots, (c) The vision projection errors under 16-shot setting, (d) The language projection errors under 16-shot setting.**

and ImageNet-R [14], to assess the out-of-distribution generalization ability of methods, followed by [52].

**Implementation Details.** Our experiments are conducted using pre-trained CLIP [35], where the visual encoder is adopted as ResNet-50 (RN50), and the textual encoder is a Transformer. The weights of these encoders remain frozen during both training and inference stages. We follow the previous works [51] to set $K =$ 1, 2, 4, 8, and 16, respectively, and follow the data preprocessing protocol in CLIP, including random cropping, resizing, and random horizontal flip. To incorporate textual information, we insert the class names into the standard templates, (*e.g.*, "a photo of [class name]"). In comparisons, we evaluate our method against LFA [32], Tip [51], and APE [55]. To ensure a fair comparison, we use the same classification approach as these methods, when comparing with APE, we utilize the textual prompt provided in CuPL [34].

**Training Details.** When compared with LFA [32], our approach follows the same fine-tuning stage described in [32], using AdamW[18] optimizer with a cosine scheduler [32], a learning rate of 5e-4 and a weight decay of 5e-4, for 50-200 iterations, and a cosine scheduler.

## 4.2 Ablation Study

In this section, we use RN50 on ImageNet[5] to evaluate the contributions and effectiveness of different components in our approach.

*4.2.1 The selection of local image features (RQ1).* The selected regions of the local image features are crucial in our method, as they determine which salient regions of the image contribute to the projection subspace. These selected local features should contain

Table 1: **The classification accuracy (%) with different projection operations under different shot settings.**

| $P_{\text{vis}}$ | $P_{\text{tex}}^i$ | $K=16$ | $K=8$ | $K=4$ | $K=2$ | $K=1$ |
|---|---|---|---|---|---|---|
| ✗ | ✗ | | | 58.83 (CLIP) | | |
| ✗ | ✔ | 61.89 | 61.83 | 61.67 | 61.48 | 61.32 |
| ✔ | ✗ | 62.44 | 62.36 | 62.14 | 61.93 | 61.95 |
| ✔ | ✔ | **64.34** | **63.65** | **63.09** | **62.53** | **61.95** |

Table 2: **The results of different visual encoders on ImageNet under a 16-shot setting. G.A. stands for GraphAdapter.**

| Models | RN50 | RN101 | ViT-B/32 | ViT-B/16 |
|---|---|---|---|---|
| CoOp (CVPR22) | 62.95 | 66.60 | 66.85 | 71.92 |
| TaskRes (CVPR23) | 64.75 | 67.70 | 68.20 | 73.07 |
| G.A. (NeurIPS23) | 64.94 | 67.87 | 68.47 | 73.40 |
| Tip (ECCV22) | 62.03 | 65.19 | 65.87 | 70.82 |
| Tip + **SSP** | 62.75(+0.72) | 65.73(+0.54) | 66.56(+0.69) | 71.56(+0.74) |
| LFA (ICCV23) | 63.65 | 67.16 | 67.63 | 72.61 |
| LFA + **SSP** | 64.34(+0.69) | 68.03(+0.87) | 68.17(+0.54) | 73.04(+0.43) |
| APE (ICCV23) | 66.02 | 69.48 | 69.31 | 74.27 |
| APE + **SSP** | **66.51**(+0.49) | **69.83**(+0.35) | **69.84**(+0.53) | **74.79**(+0.52) |

discriminative patterns relevant to the category while excluding irrelevant areas. To identify the optimal number of regions for vision subspace and language subspace, we vary the number of $Q$ and $C$ from 15 to 49, respectively. The results are depicted in Figure 5(a) and Figure 5(b). We observe that as $Q$ and $C$ increase, the averaged accuracy initially increases and then decreases. The highest accuracy is achieved in the range of 35-45. Furthermore, we analyze the disparity between the selected regions for image and text, respectively. To achieve this, we calculate the similarity between the text features and the selected regions based on image features, as well as the similarity between the image features and the selected regions based on text features. The results are shown in Figure 5(c) and Figure 5(d). In Figure 5(c), as the regions become more complete (blue curves), the similarity between the selected region features and the text label feature (red curve) decreases. Similarly, in Figure 5(d), as the number of selected regions increases, the similarity between the selected region features and image features (blue curve) also decreases. These observations and results indicate that the responded regions for image and text features are different, and selections should be performed separately for each of them.

*4.2.2 The effects of subspace construction (RQ2).* In this experiment, we investigate the effects of using different numbers of singular vectors in the construction of the vision projection matrix ($P_{\text{vis}}$) and language projection matrix ($P_{\text{tex}}^i$). Considering a feature dimension of $d = 1024$, we vary the number of singular vectors from 550 to 1000. The results are presented in Figure 6(a) and Figure 6(b). Firstly, we observe that using a small number of singular vectors in both vision and language subspace constructions leads to poor classification performance. To be specific, for the vision subspace, when the number exceeds 700, the results converge across all shot settings. This indicates that using 700 singular vectors is sufficient to span the entire vision subspace. In the case of the language subspace, we observe that the projection operation brings noticeable

Table 3: **The classification accuracy (%) comparison on few-shot learning, *i.e.*, 1-/2-/4-/6-/8-/16-shot, across 11 datasets. The results for LFA, Tip, and APE from our implementation by open public project, and the datasets include F102.(Flowers102), Euro(EuroSAT), F101.(Food101), SUN.(SUN397), C101.(Caltech101), UCF.(UFC101), and ImgN.(ImageNet).**

| Method | Pets | F102. | FGVC | DTD | Euro. | Cars | F101. | SUN. | C101. | UCF. | ImgN. | Avg. | |
|---|---|---|---|---|---|---|---|---|---|---|---|---|---|
| CLIP | 85.77 | 66.14 | 17.28 | 42.32 | 37.56 | 55.61 | 77.31 | 58.52 | 86.29 | 61.46 | 58.18 | 58.77 | |
| *16-shot* | | | | | | | | | | | | | |
| LFA (ICCV23) | **86.75** | 94.56 | **35.86** | 66.35 | 84.13 | **73.58** | 76.32 | **71.32** | 92.68 | 77.00 | 63.65 | 74.75 | |
| LFA + **SSP** | 86.40 | **95.13** | 35.10 | **67.85** | **84.83** | 73.08 | **77.79** | 69.95 | **93.35** | **78.11** | **64.34** | **75.08** | ↑0.34 |
| Tip (ECCV22) | 88.10 | 89.93 | 29.88 | 60.70 | 70.59 | 66.61 | 77.88 | 66.82 | 90.63 | 70.68 | 62.03 | 70.35 | |
| Tip + **SSP** | **88.83** | **90.62** | **30.18** | **62.23** | **73.62** | **67.19** | **77.94** | **67.01** | **91.56** | **70.95** | **62.75** | **71.17** | ↑0.82 |
| APE (ICCV23) | 87.33 | 91.19 | 32.46 | 65.78 | 77.79 | 70.36 | 78.44 | 68.94 | 91.97 | 76.74 | 63.02 | 73.09 | |
| APE + **SSP** | **88.12** | **91.51** | **32.79** | **67.91** | **78.33** | **70.77** | **78.51** | **69.14** | **92.05** | **78.46** | **63.33** | **73.72** | ↑0.63 |
| *8-shot* | | | | | | | | | | | | | |
| LFA (ICCV23) | 84.63 | 91.80 | 29.40 | 59.57 | 76.54 | 67.79 | 76.4 | 69.88 | 91.36 | 74.09 | 61.38 | 71.17 | |
| LFA + **SSP** | **84.96** | **92.89** | **30.18** | **62.00** | **78.52** | **69.58** | **77.58** | **69.95** | **91.85** | **75.05** | **62.65** | **72.29** | ↑1.12 |
| Tip (ECCV22) | 86.94 | 88.23 | 25.53 | 58.39 | 67.95 | 63.06 | 77.69 | 65.58 | 89.94 | 68.44 | 61.44 | 68.47 | |
| Tip + **SSP** | **87.19** | **88.63** | **27.78** | **58.98** | **72.28** | **63.89** | **77.75** | **65.68** | **90.91** | **69.28** | **62.22** | **69.51** | ↑1.04 |
| APE (ICCV23) | 86.97 | 90.78 | 28.38 | 63.65 | 75.04 | 65.86 | 77.71 | **67.90** | 91.60 | 70.34 | 62.63 | 70.99 | |
| APE + **SSP** | **87.35** | **91.03** | **28.86** | **65.60** | **75.16** | **67.28** | **77.88** | 67.77 | **91.72** | **72.67** | **62.64** | **71.63** | ↑0.64 |
| *4-shot* | | | | | | | | | | | | | |
| LFA (ICCV23) | 83.51 | 89.40 | 24.39 | 55.14 | 70.74 | 63.19 | **77.83** | 67.71 | 89.86 | 69.18 | 57.80 | 68.07 | |
| LFA + **SSP** | **84.06** | **89.65** | **26.88** | **57.86** | **72.31** | **63.70** | 77.60 | **68.65** | **90.87** | **71.45** | **59.91** | **69.36** | ↑1.29 |
| Tip (ECCV22) | 86.48 | 83.80 | 22.11 | 53.90 | 65.54 | 61.23 | 77.52 | 64.23 | 89.09 | 66.19 | 61.00 | 66.46 | |
| Tip + **SSP** | **86.81** | **84.13** | **23.67** | **54.79** | **67.21** | **61.47** | **77.64** | **64.27** | **90.14** | **67.80** | **61.98** | **67.26** | ↑0.80 |
| APE (ICCV23) | 85.72 | 87.66 | **25.14** | 60.46 | 73.48 | 65.19 | 77.31 | 66.87 | 91.56 | 69.15 | 62.46 | 69.55 | |
| APE + **SSP** | **86.24** | **87.86** | 24.92 | **61.35** | **74.35** | **65.38** | **77.58** | **66.95** | **91.65** | **70.10** | **62.53** | **69.90** | ↑0.36 |
| *2-shot* | | | | | | | | | | | | | |
| LFA (ICCV23) | 81.60 | 81.00 | 19.38 | 49.29 | 61.27 | 56.51 | 64.58 | 61.93 | 88.76 | 65.93 | 55.18 | 62.31 | |
| LFA + **SSP** | **82.09** | **82.26** | **22.77** | **54.37** | **63.3** | **58.81** | **65.39** | **63.19** | **89.13** | **67.09** | **57.82** | **64.20** | ↑1.89 |
| Tip (ECCV22) | 86.92 | 79.01 | 21.21 | 49.59 | 61.42 | 58.00 | 77.50 | 62.72 | 88.64 | 64.76 | 60.95 | 64.61 | |
| Tip + **SSP** | **87.03** | **79.5** | **22.71** | **50.77** | **62.36** | **59.11** | **77.62** | **62.84** | **89.01** | **66.09** | **61.82** | **65.35** | ↑0.74 |
| APE (ICCV23) | 85.20 | 83.68 | 23.55 | 54.67 | 71.89 | 61.25 | **77.62** | 65.94 | 89.94 | 66.14 | 62.38 | 67.48 | |
| APE + **SSP** | **86.07** | **83.80** | **23.64** | **57.57** | **72.37** | **61.96** | 77.27 | **66.30** | **90.63** | **66.43** | **62.41** | **68.04** | ↑0.61 |
| *1-shot* | | | | | | | | | | | | | |
| LFA (ICCV23) | 79.61 | 76.00 | 16.26 | 45.09 | 60.10 | 50.81 | 77.29 | 58.55 | 85.19 | 59.00 | 52.46 | 60.03 | |
| LFA + **SSP** | **82.45** | **77.18** | **19.68** | **51.65** | **60.94** | **56.09** | **77.30** | **61.87** | **87.87** | **66.59** | **56.11** | **63.43** | ↑3.40 |
| Tip (ECCV22) | 86.02 | 73.12 | 18.96 | 46.10 | 54.41 | 57.37 | 77.42 | 61.31 | 87.06 | 62.75 | 60.69 | 62.29 | |
| Tip + **SSP** | **86.32** | **76.05** | **19.74** | **46.81** | **59.17** | **57.60** | **77.58** | **61.49** | **88.76** | **63.02** | **61.71** | **63.48** | ↑1.19 |
| APE (ICCV23) | 85.04 | **79.98** | **21.03** | 54.31 | 67.59 | 60.25 | 77.07 | **64.55** | 89.66 | **63.36** | 62.04 | 65.90 | |
| APE + **SSP** | **85.34** | 79.51 | 20.64 | **54.73** | **68.63** | **60.29** | **77.22** | 64.36 | **90.26** | 63.34 | **62.05** | **66.03** | ↑0.14 |

Table 4: **The classification accuracy (%) comparison on out-of-distribution test data under 16-shot setting.**

| Method | Source | Target | | | | | | |
|---|---|---|---|---|---|---|---|---|
| | ImageNet | ImageNet-A | ImageNet-V2 | ImageNet-R | ImageNet-Sketch | Avg. | OOD Avg. | |
| CLIP | 58.16 | 21.83 | 51.41 | 56.15 | 33.37 | 44.18 | 40.69 | |
| CoOp (IJCV22) | 63.33 | 23.06 | 55.40 | 56.60 | 34.67 | 46.61 | 42.43 | |
| CoOpOp (CVPR22) | 62.81 | 23.32 | 55.72 | 57.74 | 34.48 | 46.81 | 42.82 | |
| LFA (ICCV23) | 63.88 | 24.31 | 55.79 | 58.13 | 34.37 | 47.29 | 43.15 | |
| **Ours** | **64.34** | **24.53** | **56.04** | **58.96** | **34.58** | **47.69** | **43.53** | ↑0.38 |

improvements, particularly in low-shot settings. When the number of singular vectors ranges from 900 to 1000, the accuracy converges across all settings, suggesting that using 900-1000 singular vectors can span the entire language subspace. Additionally, we analyze the

Table 5: **A comparison of the training time and parameters utilized by various methods and our SSP on ImageNet, using RN50 as the visual encoder on A100 GPU, under a 16-shot setting. The terms *Vis. Proj.* and *Lag. Proj.* denote Vision Projector and Language Projector, respectively.**

| Method | Time | Params. | Acc. |
|---|---|---|---|
| **SSP**(only *Vis. Proj.*) | 4.4s | - | 61.84 |
| **SSP**(only *Lag. Proj.*) | 2Min1s | - | **62.44** |
| LFA | 3Min9s | 1.05 M | 63.65 |
| +**SSP** | 4Min5s | 1.05 M | **64.34**(+0.69) |
| Tip | 5Min10s | - | 62.03 |
| +**SSP** | 6Min15s | - | **62.75**(+0.72) |
| APE | 5Min50s | - | 63.02 |
| +**SSP** | 6Min50s | - | **63.33**(+0.31) |

projection error for image features $(I - P_{\text{vis}})f_{\text{test}}$ and text features $(I - P_{\text{tex}}^i)f_{\text{test}}$ to support our analysis. The results are shown in Figure 6(c) and Figure 6(d). We find that the vision subspace projection error decreases significantly when the number of singular vectors exceeds 700, while the language subspace projection error gradually decreases. However, when the number exceeds 1000 for the language subspace, the projection error decreases to almost zero. This is because the projection matrix ($P_{\text{tex}}^i \approx I$) is nearly an identity matrix, resulting in an invalid projection operation ($f_{\text{test}} = I \cdot f_{\text{test}}$) and a projection error close to zero ($(I - P_{\text{tex}}^i)f_{\text{test}} \approx 0$). Based on the above analysis, we choose to employ 900 singular vectors for both constructing the vision subspace and the language subspace.

*4.2.3 The effects of vision and language projector (RQ3).* In this ablation experiment, we aim to demonstrate the impact of vision projector and language projector individually. The performance outcomes across different shot settings are summarized in Table 1. The results indicate that both vision subspace projection and language subspace projection lead to improvements in classification performance compared to the baseline (CLIP). Specifically, under $K = 16$, vision projection yields a 3.61% accuracy improvement, while language projection achieves a 3.01% accuracy improvement. Furthermore, it is worth noting that combining both projection operations yields the best performance, as indicated by the last row of Table 1. This outcome validates the effectiveness of utilizing both vision and language subspace projections in our method.

## 4.3 Performance Comparisons (RQ4)

We conducted a comprehensive comparison of our method with the latest approaches, including CoOp [53], TaskRes [46], GraphAdapter [22], LFA [32], APE [55], and Tip [51], across 11 image classification datasets. We first perform experiments using different visual encoders, including RN50, RN101, ViT-B/32, and ViT-B/16, on the ImageNet with a 16-shot setting. The results, as summarized in Table 2, indicate that our SSP method leads to performance improvements compared to Tip, LFA, and APE. Notably, when built upon the APE method, our SSP outperforms all other methods. Furthermore, we extended our comparisons to evaluate our method across different shot settings on various datasets, with the results outlined in Table 3. While our method can not achieve the highest performance

on some individual datasets, it consistently outperforms the other methods when considering the average accuracy across all datasets. This demonstrates the robustness of our SSP method. Specifically, compared to LFA, our method achieves a significant average accuracy improvement of 3.4% in the 1-shot setting. Furthermore, even when compared to APE, our method consistently exhibits slightly better performance across all shot settings. These results emphasize the effectiveness of our approach in improving few-shot image classification performance. For further comparisons, we also evaluated our method against Tip-F [51] and APE-T [55]. The detailed results of these comparisons can be found in the Supplementary Material.

## 4.4 Domain Generalization Comparisons (RQ5)

To assess the generalization ability of our method, we conducted an experiment using the ImageNet dataset as the source dataset, which provides 16-shot training images for each category. We then tested our method on four different variants of the ImageNet dataset: ImageNetV2 [36], ImageNet-Sketch [42], ImageNet-A[15], and ImageNet-R [14]. These test datasets share the same class labels as ImageNet but exhibit semantic gaps. The classification results of our method on these test datasets are summarized in Table 4. Our method achieved a significant improvement of 0.73% over the LFA method specifically on the ImageNet-R dataset. Furthermore, our method demonstrated an average performance improvement of 0.38% across all out-of-distribution (OOD) datasets. These results demonstrate our SSP is robust in domain adaptation.

## 4.5 Analysis of Computation Efficiency (RQ6).

We conducted a computation efficiency analysis by comparing the computation time and parameters between our SSP and other approaches on the ImageNet dataset, under the 16-shot setting. The results are summarized in Table 5. It's worth noting that our SSP does not introduce any learnable parameters and primarily incurs computation time during matrix operations. To align image features, SSP requires a single SVD operation, typically taking a few seconds that could be ignored. As for text features alignment, the number of SVD operations depends on the total categories in the dataset (*e.g.*, 1000 categories in ImageNet). When combined with LFA, Tip, or APE, our SSP needs nearly an extra 1 minute computation time, but it consistently improves the classification accuracy, particularly by 0.72% over Tip.

## 5 CONCLUSION

In this paper, we have proposed a method named **S**elective Vision-Language **S**ubspace **P**rojection to align the different modality features extracted for pre-trained CLIP through subspace projection. This alignment strengthens the generalization ability of CLIP in the few-shot scenarios. Our SSP is seamlessly integrated into various classification frameworks and does not introduce any additional learnable parameters. The extensive experiments validate the effectiveness of our proposed method. However, it's worth noting that the performance improvements are limited on the ImageNet dataset. In our future work, we plan to explore more general alignment techniques by incorporating other pre-trained models, such as large language models, diffusion models, *etc*.

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
