# OpenReview forum: "Selective Vision-Language Subspace Projection for Few-shot CLIP"
_acmmm.org/ACMMM/2024/Conference — MM2024 Oral_

### Official Review · Reviewer_yY8r · 2024-05-16

**Rating:** 5
**Confidence:** 3

**Summary:**

This paper proposes a selective vision-language subspace projection method to enhance the alignment between image-text pairs by utilizing local image features. The proposed approach reduces the modality gaps in CLIP’s encoded features, which improves the CLIP’s few shot generalization ability. Sufficient experimental evaluations are provided.

**Strengths:**

1. The paper summarized a comprehensive review of the current progress of related work and pointed out the limitations.
2. The method has achieved significant improvements in experiments.
3. This paper is well organized and written.

**Limitations:**

Some concerns and suggestions to improve this work are as follows:
1. The introduction and related work sections would benefit from including a more comprehensive set of references, particularly few-shot clip studies [1, 2], to provide a more thorough overview of the existing literature in the field.
[1] Domain Aligned CLIP for Few-shot Classification, Workshop on Applications of Computer Vision, 2024.
[2] Contrastive knowledge-augmented meta-learning for few-shot classification, Workshop on Applications of Computer Vision, 2023.
2. The main idea is easy to follow. The author proposes to use the regions of the local image features to align the image and text features via subspace projection. Could you please give some qualitative analysis to explain this?
3. It is recommended that the authors check any potential citation errors and enhance the manuscript's overall quality.

**Suitability:**

3

---

### Official Review · Reviewer_mxke · 2024-05-19

**Rating:** 5
**Confidence:** 4

**Summary:**

This paper proposes a training-free method called Selective Vision-Language Subspace Projection (SSP) method for few-shot CLIP. The approach combines two parallel modules: a vision projector and a language projector to achieve alignment. Experimental results on several benchmark datasets are reported.

**Strengths:**

A novel problem definition and a well motivated solution.
The paper is overall well-written and easy to follow.
Both quantitative and qualitative interpretations are discussed in detail.

**Limitations:**

Some concerns and suggestions to improve this work are as follows:
1. The methods mentioned in the abstract are unclear, and there is also a lack of corresponding explanation in Section I, e.g. “However, most existing methods overlook modality gaps in CLIP’s encoded features, which is shown as the text and image features lie far apart from each other, resulting in limited classification performance”.
2. The summary of the contributions in the manuscript is not sufficiently clear. It would be beneficial to explicitly outline the novel aspects and specific contributions of the proposed method or findings.
3. The paper lacks specific details regarding the practical implementation.
4. Reference formatting is inconsistent. Please carefully revise the references, ensuring consistent use of either full names or abbreviations for journal and conference names throughout the manuscript.

**Suitability:**

2

---

### Official Review · Reviewer_CbuB · 2024-05-22

**Rating:** 4
**Confidence:** 2

**Summary:**

The paper introduces a technique titled Selective Vision-Language Subspace Projection, which aligns the multimodal features derived from the pre-trained CLIP model via a subspace projection approach. This alignment strengthens the generalization ability of CLIP in the few-shot scenarios. The proposed module can be effortlessly incorporated into a variety of classification frameworks, adding no extra trainable parameters. Comprehensive experimental results validate the effectiveness of the proposed method across diverse benchmarks.

**Strengths:**

- The proposed method has good generality and has demonstrated effectiveness on multiple datasets.
- The paper is well-written and easy to follow.
- The experiments are comprehensive, and the results demonstrate that the proposed method can be effectively integrated into various classification frameworks, leading to performance improvements.

**Limitations:**

- When constructing the language subspaces, local image features extracted by the visual encoder are used. Can we still call them language subspaces? This somewhat undermines the persuasiveness of Figure 1 and Figure A.
- According to Figure 2 (middle), the text features focus on unrelated regions, and these regions are selected via Top-C. Why can language subspaces constructed with the assistance of local image features from these unrelated areas achieve the results shown in Figure 2 (bottom)?
- In Table 2 and Table 5, the performances of APE and APE+SSP are inconsistent.

**Suitability:**

3

---

### Official Review · Reviewer_hiNC · 2024-05-24

**Rating:** 4
**Confidence:** 2

**Summary:**

This manuscript introduces Selective Vision-Language Subspace Projection (SSP) to solve the alignment issue in the vision and language feature spaces of image-text pairs. The proposed SSP utilizes vision and language projectors to align the image and text features via subspace projection.

**Strengths:**

1. This manuscript focuses on an important alignment issue in existing CLIP-based methods and introduces an effective model by incorporating local image features and subspace projector to tackle the issue.
2. The experimental results demonstrate the effectiveness of the proposed approach. And the visualization examples also show a better concentration on important regions.
3. The ablation experiments presented in the manuscript are comprehensive.

**Limitations:**

1. Some unclear explanation: the manuscript lacks some explanation about the details of the methodology. For instance,
  (1) what are the candidate regions from which the selected regions are chosen? Could the authors provide more details about the generation process of such regions?
  (2) According to Figure 1, the proposed method can largely alleviate the misalignment issue, but the final results only show a relatively limited improvement. Besides, the original CLIP-based methods can also achieve good results with such large modality gap. It seems that removing modality gap brings limited improvement. Could the authors give an explanation.

**Suitability:**

2

---

### Meta-Review · Area_Chair_QJcf · 2024-07-05

**Recommendation:** Accept (Oral)
**Confidence:** 5

**Metareview:**

The manuscript introduces the so-called Selective Vision-Language Subspace Projection (SSP) method to address modality gaps in vision-language models like CLIP. The main idea is to utilize local image features for better text-image alignment. The proposed SSP framework include a vision projector and a language projector, which are used to enhance alignment through training-free matrix calculations and integrate seamlessly into existing few-shot learning frameworks. Extensive experiments demonstrate SSP's superior performance over state-of-the-art methods.

The manuscript received two borderline accept and two accept ratings. Strengths include the innovative, practical approach, comprehensive evaluation, and code availability, while weaknesses involve implementation details and concerns from borderline reviews.  The authors' rebuttals partially addressed these weakness. After considering these factors, I recommend accepting this paper. Authors should add detailed implementation explanations, address reviewer concerns, and highlight future research directions in the final version.